# Functional basis of electron transport within photosynthetic complex I

Katherine H. Richardson[1,2], John J. Wright [1,3], Mantas Šimėnas[4], Jacqueline Thiemann [5], Ana M. Esteves[1], Gemma McGuire[1,2], William K. Myers [6], John J. L. Morton [4,7], Michael Hippler [8,9], Marc M. Nowaczyk [5], Guy T. Hanke [1✉] & Maxie M. Roessler [2✉]

Photosynthesis and respiration rely upon a proton gradient to produce ATP. In photosynthesis, the Respiratory Complex I homologue, Photosynthetic Complex I (PS-CI) is proposed to couple ferredoxin oxidation and plastoquinone reduction to proton pumping across thylakoid membranes. However, little is known about the PS-CI molecular mechanism and attempts to understand its function have previously been frustrated by its large size and high lability. Here, we overcome these challenges by pushing the limits in sample size and spectroscopic sensitivity, to determine arguably the most important property of any electron transport enzyme – the reduction potentials of its cofactors, in this case the iron-sulphur clusters of PS-CI (N0, N1 and N2), and unambiguously assign them to the structure using double electron-electron resonance. We have thus determined the bioenergetics of the electron transfer relay and provide insight into the mechanism of PS-CI, laying the foundations for understanding of how this important bioenergetic complex functions.

[1] School of Biological and Chemical Sciences, Queen Mary University of London, London, UK. [2] Department of Chemistry, Imperial College London, Molecular Sciences Research Hub, London, UK. [3] Medical Research Council Mitochondrial Biology Unit, Wellcome Trust/MRC Building, Cambridge, UK. [4] London Centre for Nanotechnology, University College London, London, UK. [5] Plant Biochemistry, Faculty of Biology and Biotechnology, Ruhr University Bochum, Bochum, Germany. [6] Inorganic Chemistry, University of Oxford, Oxford, UK. [7] Department of Electronic & Electrical Engineering, UCL, London, UK. [8] Institute of Plant Biology and Biotechnology, University of Münster, Münster, Germany. [9] Institute of Plant Science and Resources, Okayama University, Kurashiki, Japan. ✉email: g.hanke@qmul.ac.uk; m.roessler@imperial.ac.uk

The majority of life on earth is dependent on photosynthesis, which uses light energy to generate potential energy in the form of a proton gradient. The transfer of electrons is coupled to the movement of protons across a membrane by a set of exquisitely efficient molecular machines. Of these, the photosystems (PSII and PSI) and the cytochrome $b_6f$ complex have been well characterised, but until recently, little information was available about an additional proton pump, photosynthetic complex I (PS-CI, previously known as NDH-1). PS-CI is a key component of cyclic electron flow (CEF) in cyanobacteria and plants[1–4]. Photosynthetic organisms utilise CEF around photosystem I to increase the transmembrane proton gradient and thus ATP production, to meet the ATP:NADPH ratio required for $CO_2$ fixation[5,6]. In addition to its role in cyanobacteria, it has also been shown that PS-CI is important in many higher plants, including several crops. This is especially the case when ATP demands are high, such as when performing C4 type photosynthesis[7], or sustaining growth under low light or other stresses[8]. In this way, PS-CI is critical to yield in some crops[9]. PS-CI accepts electrons from the terminal electron acceptor of PSI, ferredoxin (Fd) and reduces plastoquinone (PQ), coupling this electron transfer to the pumping of protons (Fig. 1a). Understanding the intricate electron transfer process that provides the free energy for proton translocation is not only of fundamental interest but important because it can fine-tune the redox state of the compartment or cell under stress[10].

PS-CI was first identified as a homologue to respiratory complex I (R-CI)[11,12]. Recent cryo-EM structures have confirmed that PS-CI is a large multisubunit membrane protein with L-shaped architecture[13,14]. It comprises 11 core subunits and seven oxygenic photosynthesis-specific subunits (OPS) which are found in both hydrophobic and hydrophilic domains. The hydrophobic arm has four Mrp (Multiple-resistance-and-pH)-like $Na^+/H^+$ antiporters which likely translocate protons (Fig. 1a)[2,15]. Although the core hydrophilic subunits are very similar in structure to their R-CI counterparts (Fig. 1b), the hydrophilic domain is truncated by three subunits, including the NADH binding domain.

Electron transfer from NADH to ubiquinone through R-CI has been extensively studied; a non-covalently bound flavin mononucleotide transfers the two electrons from NADH singly down a series of seven iron-sulfur (FeS) clusters to the ubiquinone binding site (Fig. 1b)[16,17]. Although mechanisms have been suggested, how the reduction of ubiquinone is coupled to proton pumping is not yet fully understood[18–20]. Electron paramagnetic resonance (EPR) spectroscopy has been a powerful technique to uncover critical information about the molecular environment of the FeS clusters and the movement of electrons through R-CI. When reduced by its native substrate NADH, up to five of the clusters can be observed in the EPR spectrum, with the other clusters remaining oxidised and therefore EPR silent[21]. The FeS cluster EPR signals have been named in order of their relaxation times (N1b > N2 > N3 > N4 > N5, where N5 relaxes the fastest and where N1b is a [2Fe-2S] cluster) and assigned to those identified in the structure[22,23] (Fig. 1b), revealing a 'rollercoaster' of alternating high and low reduction potential clusters, with the terminal [4Fe-4S] cluster N2 transferring electrons to ubiquinone[21,23–25]. Notably, although exact values vary and depend on pH, the N2 cluster is consistently more positive in reduction potential than the other FeS clusters[18,26,27]. In R-CI, cluster N2 is therefore postulated to act as an electron sink and may avert reverse electron transfer under physiological conditions, preventing backflow to $O_2$ that would generate dangerous superoxide radicals via the reduced flavin[28].

PS-CI accepts electrons from a one-electron donor, Fd, and is regulated by the OPS subunits unique to the photosynthetic complex[13,29]. Until recently, difficulties in purifying sufficient functional PS-CI have inhibited attempts to understand its function, and our knowledge has been limited to its subunit composition[1,10,30], the phenotype of mutants and studies of its regulation using in vivo, or semi in vitro systems[31–33]. Thus, although several structures have recently been published due to advances in cryo-electron microscopy[13,14,29], in contrast to R-CI, there is no experimental information about the molecular mechanism of electron transfer within PS-CI. In R-CI, EPR spectroscopic data, in particular information on FeS cluster

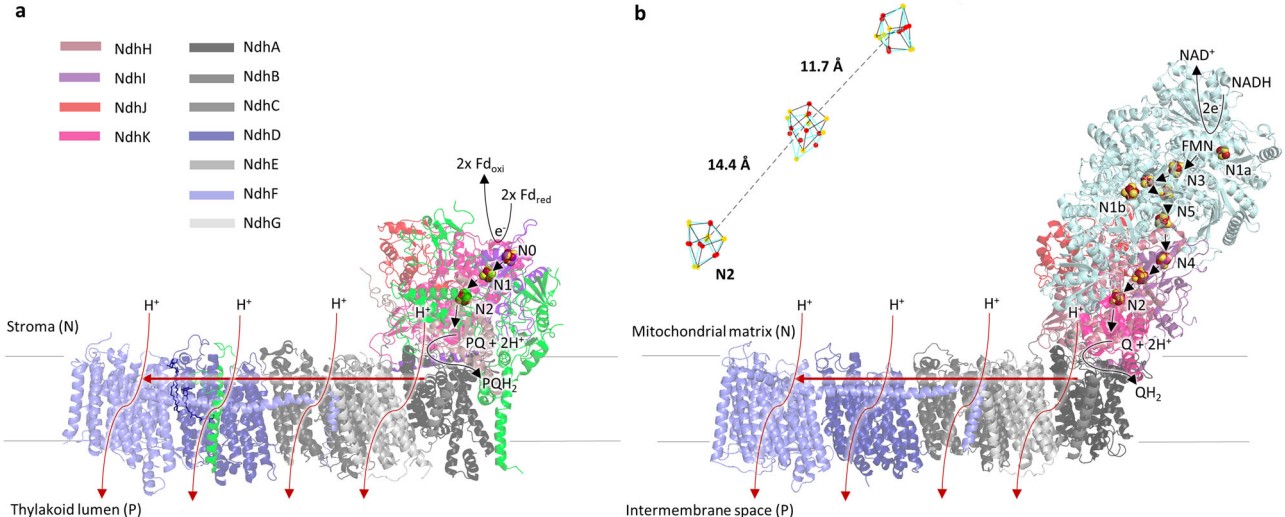

**Fig. 1 Structural and functional comparison of PS-CI and R-CI. a** Structure and proposed catalysis of *T. elongatus* PS-CI (Photosynthetic complex I) (PDB: 6HUM). The 11 core subunits are coloured as indicated; the seven oxygenic photosynthesis-specific (OPS) subunits are in green; reactions are shown schematically. Electron transfer from the donor Fd to the acceptor PQ is indicated by black arrows. The putative movement of protons across (horizontal arrow) and through the membrane domain are indicated by red arrows. **b** Structure and proposed catalysis of *T. thermophilus* R-CI (PDB: 4HEA). The analogous subunits to PS-CI are coloured using the same key as in **a**, all other subunits are pale blue; reactions are shown as in **a**. FeS clusters in R-CI are labelled according to their EPR signals[73]. **b** Inset FeS clusters (Fe in red and S in yellow) of PS-CI (black, PDB: 6HUM) superimposed with structurally equivalent clusters in R-CI (Respiratory complex I) (cyan, PDB: 4HEA). Centre-to-centre PS-CI FeS cluster distances are labelled.

reduction potentials[18,25], preceded structural information by several decades, with pulse EPR later enabling a definitive assignment of the cluster properties to their spatial location in the electron transfer chain[34]. Although the presence of PS-CI was discovered in 1998[35,36], there is—perhaps surprisingly—no information on the reduction potentials of the electron transfer centres. Without reduction potentials, it is difficult to even formulate a hypothesis on how this molecular machine works. The lack of this fundamental parameter for any electron transfer enzyme may be due to experimental bottlenecks, because EPR-based potentiometric titrations and detailed pulse EPR measurements typically require very large amounts of enzyme. In addition, the high magnetic anisotropy and extensive spin delocalisation of FeS clusters, whose EPR signals all overlap, make them one of the most challenging paramagnetic centres to work with.

Here, we overcome these experimental bottlenecks and not only determine the reduction potentials of the FeS clusters, but also assign their position in the electron transfer chain. We characterise the reduced FeS clusters of PS-CI from two strains of cyanobacteria using a combination of pulsed and continuous wave (CW) EPR spectroscopic methods. We determine the $g$ values for all clusters and the reduction potentials of the two fully reducible clusters. Moreover, we provide a conclusive assignment of thermodynamic properties to structurally defined FeS counterparts, giving insight into the functional mechanism of electron transfer in this crucial enzyme, placing it into the redox map of photosynthesis, and providing an essential foundation for future work on PS-CI.

## Results

### Identification of three distinct reduced [4Fe-4S] clusters in PS-CI by EPR. To study the FeS clusters of PS-CI, the complex was

purified from *Thermosynechococcus elongatus* using a native His-tag on NdhF1[14], and from *Synechocystis* sp PCC6803 with a recombinant His-tag on *Ndh-J*. The presence of the subunits was confirmed by proteomics (Supplementary Tables 1, 2). The isolated complexes were reduced using sodium dithionite and suggest the presence of three reduced FeS cluster CW EPR signals (Fig. 2a). Our recently reported high sensitivity EPR setup with a low-noise cryogenic preamplifier[37] was employed to distinguish the overlapping FeS signals by performing different pulsed EPR relaxation filtering experiments[38]. Relaxation filtering selectively recovers the different FeS cluster spectra based on their spin–lattice and spin–spin relaxation times (Supplementary Tables 1, 2 and Note 1). The N2 $g$ values match very well with those for R-CI for both species of cyanobacteria[24,39,40] (Supplementary Tables 3, 4). Consistent with what is observed for R-CI, this FeS signal is observed at relatively high temperatures (20 K) and long relaxation times[38]. Given the structural and spectroscopic similarity between the species and R-CI N2, and in agreement with previous work on *T. elongatus* CI[13], we assign the N2 EPR signal to the [4Fe-4S] cluster closest to the quinone binding site. Notably, a FeS cluster with these $g$ values and with reduction potential $-270 \pm 25$ mV was previously observed in EPR spectra of chemically reduced thylakoid membranes from two species of *Nostoc*[41], however, its origin was unknown until now.

Continuous wave and pulsed EPR spectra of PS-CI in both species (Fig. 2a and Supplementary Figs. 1, 2) could be well simulated (red traces) assuming two fully reduced [4Fe-4S] clusters and a third partially reduced (approximately half of the clusters in the enzyme sample giving rise to this third set of EPR signals are reduced at this potential), which possess characteristic $g$ values (Fig. 2a). To decrease confusion with structural or spectroscopic [4Fe-4S] nomenclature for R-CI we

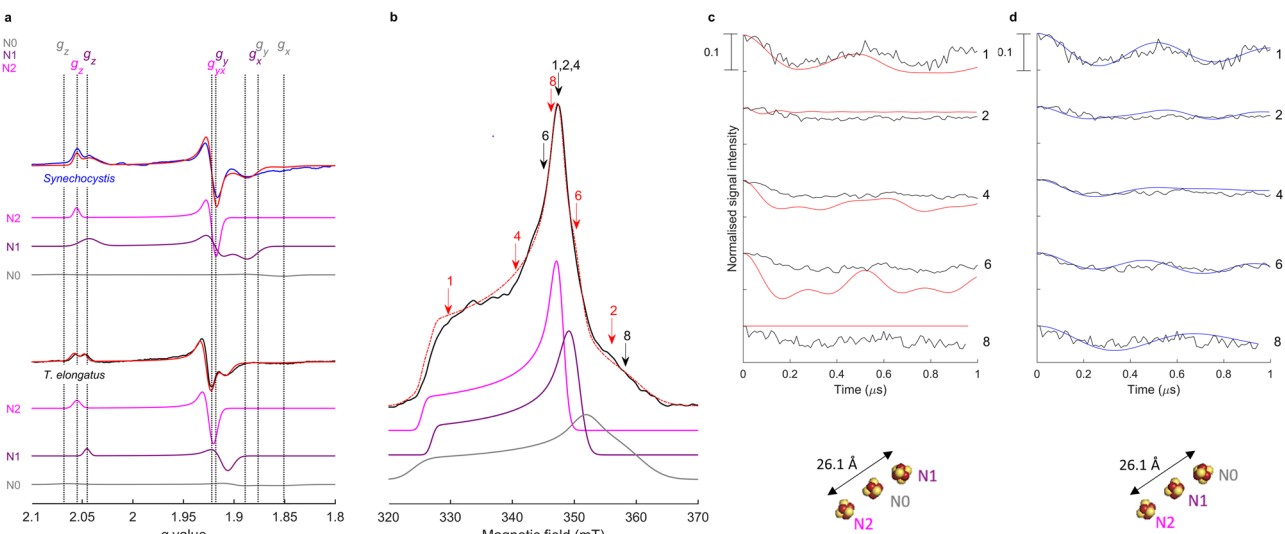

**Fig. 2 Assignment of the FeS cluster EPR signals to the structure of *T. elongatus* PS-CI using DEER spectroscopy. a** Continuous-wave EPR spectra (15 K) of the FeS clusters in sodium dithionite reduced PS-CI (Photosynthetic complex I) of *Synechocystis* (blue) and *T. elongatus* (black), 2 mW, 100 kHz modulation frequency, 7 G modulation amplitude. Simulations of the total (red) and individual N2 (pink), N1 (purple) and N0 (grey): *Synechocystis* $g$ values: N2 ($g_{x,y} = 1.922$, $g_z = 2.055$), N1 ($g_x = 1.886$, $g_y = 1.927$, $g_z = 2.043$), N0 ($g_x = 1.851$, $g_y = 1.867$, $g_z = 2.079$); *T. elongatus* $g$ values: N2 ($g_{x,y} = 1.922$, $g_z = 2.055$), N1 ($g_x = 1.907$, $g_y = 1.913$, $g_z = 2.045$), N0 ($g_x = 1.852$, $g_y = 1.899$, $g_z = 2.064$); See Supplementary Table 3 for full simulation parameters. **b** Set up of the pump pulse positions (red) and detection pulse position (black) for the corresponding DEER traces (for full experimental set up see Fig. S3) (10 K). Echo-detected field sweep of *T. elongatus* (black), the sum of simulations (red), N2 (pink), N1 (purple), N0 (grey) (N2:N1:N0 ratio 1.00:0.92:0.90); see Table S3 for simulation parameters. **c, d** *T. elongatus* orientation-selective DEER traces for the corresponding pump and probe positions (black) (10 K). The modulation depth is indicated as a scale bar. **c** Best-fit simulated DEER traces for model A, with the N1 cluster at 26.1 Å from N2 (red). **d** Best-fit simulated DEER traces for model B, with N0 at 26.1 Å from N2 (blue). Schematics of the structural models are shown below. Note that the shorter distances (i.e. dipolar coupling to the middle cluster) do not contribute to the DEER traces; see Supplementary Note 2 for details on the simulation of the DEER traces and models employed. See Fig. S4 for a complete set of the DEER traces and simulations.

refer to the remaining clusters as N1 and N0 for the reduced and partially reduced clusters, respectively (Supplementary Note 1). The g values of the second fully reduced cluster N1 are similar between the two cyanobacterial species. However, in *Synechocystis* PS-CI N1 exhibits increased broadening, likely due to increased structural variation compared to *T. elongatus*[42]. N0 appears to be only partially reduced and its EPR signal is relatively broad. Although the g values of N1 and N0 are broadly consistent with those of other R-CI clusters, they do not match any one cluster well enough to assign them on the basis of homology (Supplementary Tables 3, 4). However, the PS-CI FeS cluster g values are similar between the photosynthetic species despite a wide evolutionary distance[43]; indicating that any heterogeneity between the complexes does not have a major effect on the FeS characteristics. Deconvolution of the three overlapping FeS signals in PS-CI through relaxation filtering provides unequivocal assignment of their g values.

**Assignment of the [4Fe-4S] cluster EPR signals to the structure of PS-CI using DEER.** To assign the respective PS-CI FeS cluster EPR signals to the clusters within the structure, we used double electron-electron resonance (DEER) spectroscopy (a pulsed EPR experiment that employs two microwave frequencies)[44]. The dipolar coupling between paramagnetic centres at the 'pump' and 'probe' microwave frequencies can be measured by analysing the modulation of the DEER spectra—this coupling strength is inversely proportional to the cubic distance between the centres providing structural information about the system (Supplementary Note 2)[45]. Multiple pump/probe positions that span the entire PS-CI EPR spectrum must be collected to calculate FeS cluster interaction distances (Fig. 2b), as their highly anisotropic nature and the limited bandwidth of microwave pulses results in a partial excitation of the EPR spectrum (orientation selection). The orientation-selective DEER spectra were simulated with a custom programme adapted from one previously developed for R-CI based on a local spin model[24] (see Supplementary Note 2 and Fig. 6), taking into account the cluster positions (PDB:6HUM)[13] and our experimentally determined g values (Fig. 2c, d). With the position of N2 fixed, there are two possible models: model A, in which N2 and N1 are 26.1 Å apart, and model B in which N2 and the partially reduced cluster N0 are 26.1 Å apart (Supplementary

Fig. 4). Only model B provides a good fit at all experimental pump and probe positions, both in terms of modulation frequency and depth. This is especially apparent at field position 9, where N1 does not contribute at the detection pulse position (Fig. 2). We, therefore, assign the N0 signal to the [4Fe-4S] cluster adjacent to the Fd binding site.

**Determination of the reduction potentials of the [4Fe-4S] clusters in PS-CI.** Once the spectroscopic signatures of the clusters were assigned to structural positions, we determined the reduction potentials and therefore energetic favourability of electron transfer to and within PS-CI using small-volume potentiometric redox titrations (Fig. 3)[46]. The EPR signal intensity of each cluster at each potential was estimated based on the integration of the simulated spectra, given the FeS cluster signals overlap (Fig. 3 and Supplementary Fig. 7). The reduction potentials were estimated to be −220 and −230 mV ± 15 mV vs the standard hydrogen electrode (SHE) for N2 and N1, respectively, based on fitting the experimental data points to the one-electron Nernst equation. These values were consistent between cyanobacterial species. The reduction potentials are thus very similar and within experimental error not only between species but also between the clusters (Fig. 3). The clusters are therefore almost isopotential, meaning that electron transfer to N2 is as favourable as to N1.

The absence of a reduced N0 signal at −431 mV using measurement parameters that maximise N0 (Supplementary Fig. 8) indicates that the reduction potential must be below ~ −550 mV vs SHE. Such a low reduction potential of N0 will result in N2 and N1 being preferentially reduced, preventing the backflow of electrons to form dangerous oxygen radicals in the cytosol. This is particularly important given the unknown contribution of PS-CI to free radical production, a process that initiates multiple defence and developmental signalling cascades in photosynthetic organisms[47,48]. R-CI is a notorious generator of the superoxide radical[49,50], and it has recently been shown that blocking reverse electron transfer from the quinone site to the terminal flavin moiety prevents ROS production, protecting against cardiac ischaemia-reperfusion injury[51]. PS-CI lacks this flavin cofactor, but the terminal FeS cluster is so solvent-exposed that reverse electron transfer could also result in considerable free radical production[13,14,29].

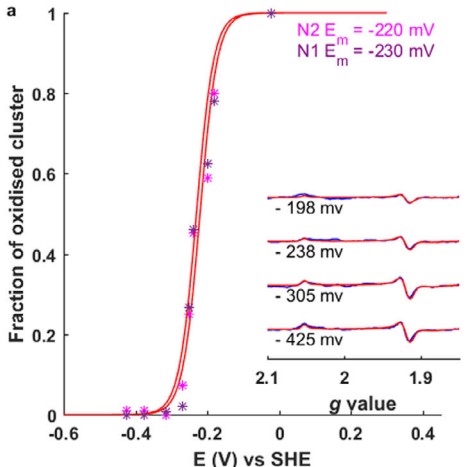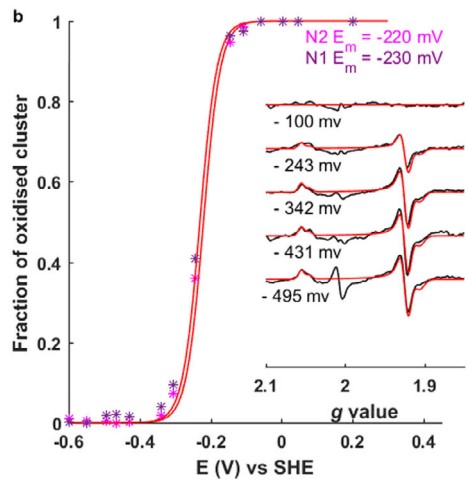

**Fig. 3 The reduction potentials of PS-CI N2 and N1.** Small-volume potentiometric titrations of PS-CI (Photosynthetic complex I) from **a** *Synechocystis* and **b** *T. elongatus*. The fraction of oxidised cluster (N2 or N1) was determined from the integration of the simulated continuous wave EPR spectra (insets, simulations in red) normalised against fully reduced N2 or N1. Data points were fitted with the one-electron Nernst equation using the indicted midpoint potentials ($E_m$). The g ~2 signal likely originates from the redox mediators required to perform the titration.

## Discussion

We were intrigued to find two clusters with equal and relatively positive $E_m$ adjacent to each other as electrostatic repulsion would suggest this to be energetically unfavourable[52]. Moreover, our results are contrary to what is observed in R-CI, where with the exception of the *E. coli* enzyme (and potentially *Thermo thermophilus* where the N2 signal is not observed) cluster N2 has the most positive reduction potential of the EPR-visible Fe-S clusters[18], with the adjacent cluster remaining oxidised upon NADH reduction[21]. Hence, the two adjacent isopotential clusters in PS-CI challenges the mechanistic principle of alternating high- and low-potential clusters in electron transfer relays[21,53–55]. The redox potentials are very similar between the two species indicating this property is conserved. The isopotential nature of the clusters suggests that reduction of cluster N2 is unlikely to be involved in the reaction that couples electron transfer and proton translocation, in line with what is known about R-CI[56,57]. On the other hand, and contrary to R-CI, N2 and N1 in PS-CI may both be electron sinks, thereby facilitating the likely rate-limiting two-electron PQ reduction required for activity, from the one-electron donor Fd.

In the cell, the very negative reduction potential of N0 would limit reduction of PS-CI until the Fd pool is in a highly reduced state (at least 100 times more abundant than PS-CI based on copy per cell estimates)[58,59], with the close proximity of Fd to N0 providing an electron tunnelling pathway from Fd to N1/N2[29,52]. It remains possible that the potential is altered by the binding of loosely associated subunits (such as NDH-V[60]) as is observed in R-CI[61], or the substrate itself, as seen in photosystem I[62,63]. It has been suggested that electron transfer can occur in reverse in PS-CI if the proton motive force is high or the PQ pool is predominantly reduced, such as under decreased $CO_2$ or fluctuating light intensities[2]. The very negative potential of N0 means that should such conditions occur, electron transport from N0 to Fd would be highly favourable, improving the capacity of Fd to effectively compete with oxygen as an oxidant and preventing oxidative damage. The reduction potentials established in this work, therefore, provide a basis for understanding under which cellular conditions the PS-CI complex is active, and how it could be intrinsically regulated by the redox status of the cell.

The mechanism of function for both R-CI and PS-CI remain a matter of proposals that are still under debate, and although that of R-CI is much better understood, it is nonetheless unclear how electron transport is coupled to proton translocation across the membrane. The reduction potentials provided in this work clarify how the electron transfer pathway functions, and establish the energy gaps between the Fd electron donor and N0, between N0 and N1/N2 and between N2 and free quinone. This provides a platform for future work, in which hypotheses can be tested regarding how electron transfer within PS-CI might be coupled to proton pumping. We have shown that overall electron transport to the quinone pool (at $+80\,mV$[64]) is energetically very favourable in PS-CI. However, there is no consensus on whether, or how, a quinone radical, which is anticipated to be $\sim 12\,\text{Å}$ from N2 by analogy with R-CI[65,66], would be stabilised in PS-CI[14,29,67]. Without accurate estimates of the reduction potential of PQ within PS-CI it is not yet possible to determine whether the coupling energy is indeed provided by quinol release, the latest theory-derived model suggested for PS-CI[68], or quinone reduction.

Here we provide insight into the basis of the electron transfer mechanism in PS-CI, placing it into the bioenergetic network of photosynthesis and CEF in cyanobacteria. By applying custom EPR instrumentation and simulating the dipolar interactions between highly delocalised and anisotropic paramagnetic species, the individual FeS cluster EPR signals and their corresponding structural position have been assigned using dilute (<15 μM) and low-volume (~10 μL) samples. The reduction potentials of the N1 and N2 clusters allow overall favourable oxidation of reduced Fd and reduction of PQ, releasing the energy required for generating the proton gradient used by ATP synthase. We reveal the isopotential nature of the non-solvent exposed clusters, providing an energetic trap when Fd is not bound and theoretically preventing non-specific reverse electron transfer. These findings produce a basis for how PS-CI works in energetic terms, with the reduction potentials providing a solid platform for future studies to solve its functional mechanism and clarify its role in photosynthetic electron transport systems.

## Methods

**Purification of photosynthetic complex I**. PS-CI was purified from *Synechocystis* sp. pcc 6803 by introducing a His-tag to the Ndh-J subunit. A synthetic DNA sequence corresponding to a fragment of the *Synechocystis* sp. PCC6803 genome stretching 400 b.p. both upstream and downstream of the *Ndh-J* gene was synthesised (GENEWIZ, Leipzig, Germany) (Supplementary Fig. 9). The additional sequence was incorporated, encoding a 6X His-tag in the frame at the C-terminal end of the gene for the NDH-J protein, connected to the protein by a Factor Xa stie. This was followed by both rrnV T1 terminator and T7Te terminator sequence. The sequence was inserted into pEX-K4, and a fragment containing the KanR gene for kanamycin resistance sub-cloned over the StuI and SacI restriction cloning sites downstream, but anticoding, to the *Ndh-J* gene. All enzymes for molecular biology were from New England Biolabs (Ipswich, MA, USA). A purified plasmid was used for the transformation of wild type *Synechocystis* sp. PCC6803, followed by selection on agar-BG11 supplemented with kanamycin at a concentration of 50 μg/mL at 30 °C under continuous illumination of 50 μE·m⁻²·s⁻¹. Large scale growth (70 L of culture) and purification were performed as described[69]. Cells were resuspended in 20 mL/L 20 mM sodium phosphate pH 7.5, 5% glycerol, 5 mM $MgCl_2$, 10 mM NaCl, 1 mM benzamidine, 1 mM aminocaproic acid and 100 μM protease inhibitor. Cells were disrupted by passing twice through a microfluidiser (30,000 psi). Lysate supernatant was centrifuged at 50,000 x *g* for 60 min at 4 °C. Membranes were solubilised in 5 mm $MgSO_4$, 20 mM MES pH 6.5, 10 mM $MgCl_2$, 10 mM $CaCl_2$, 25% glycerol (henceforth BB) with a final concentration of 1% (w/v) *n*-Dodecyl β-D-maltoside. The suspension was centrifuged at 50,000 x *g* for 30 min at 4 °C. The supernatant was passed down a 25 mL Ni-NTA column equilibrated in BB ( +10 mM imidazole, 0.03% DDM). PS-CI was eluted in BB supplemented with 200 mM imidazole and desalted by PD-10 column prior to EPR sample preparation. PS-CI was purified from *Thermosynechococcus elongatus* as previously described using a native His-tag on the Ndh-F subunit[14]. Cells were resuspended in 20 mM MES, 10 mM $MgCl_2$, 10 mM $CaCl_2$ pH 6.5 (Buffer A) and incubated with lysozyme (0.2% w/v) for 90 m. Cells were disrupted using a Parr bomb and washed in buffer A + 0.5 M mannitol. Membranes were solubilised and purified using Ni-NTA as described above. Prior to EPR sample preparation size exclusion chromatography was performed in 20 mM HEPES pH 8.0; 0.5 M mannitol; 150 mM NaCl; 0.03% (w/v) DDM. Both samples were concentrated in 100 kDa MWCO spin concentrators and PS-CI subunit presence was confirmed using mass spectrometry.

**Liquid chromatography-mass spectrometry**. In-gel digestion was carried out using sequencing-grade trypsin (Promega) according to standard protocols, without reduction and carbamidomethylation of cysteines[70]. Chromatographic separation of peptides was performed using an Ultimate 3000 RSLCnano System (Dionex, part of Thermo Fisher Scientific). The sample (3 μL) was loaded on a trapping column (C18 PepMap 100, 300 μM × 5 mm, 5 μm particle size, 100 Å pore size; Thermo Scientific) and desalted for 5 min using 2.5% acetonitrile/0.05% trifluoroacetic acid in ultrapure water at a flow rate of 10 μL/min. Then the trap column was switched in line with the separation column (Acclaim PepMap100 C18, 75 μm × 50 cm, 2 μM particle size, 100 Å pore size, Thermo Scientific). The mobile phases for peptide elution consisted of 0.1% (v/v) formic acid in ultrapure water (A) and 80% acetonitrile/0.1% (v/v) formic acid in ultrapure water (B). Peptides were eluted at a flow rate of 300 nL/min with the following gradient profile: 2.5% B over 5 min, 2.5–45% B over 40 min, 45–99% B over 5 min and 99% B over 20 min. Afterwards the column was re-equilibrated with 2.5% B for 45 min. The LC system was coupled via a nanospray source to a Q Exactive Plus mass spectrometer (Thermo Fisher Scientific). MS full scans (m/z 300−1600) were acquired in positive ion mode by FT-MS in the Orbitrap at a resolution of 70,000 (FWHM) with internal lock mass calibration on m/z 445.12003. The 12 most intense ions were fragmented with 27% normalised collision energy at a resolution of 17,500 and a maximum injection time of 50 ms. Automatic gain control (AGC) was enabled with target values of 3 × 106 and 5 × 104 for MS full scans and MS/MS, respectively. One microscan was acquired per MS/MS spectrum and the maximum fill time was 55 ms. Dynamic exclusion was enabled with an exclusion duration of 30 s and an exclusion mass width of ± 5 ppm. Ions with unassigned charge states, singly charged ions as well as ions with charge state >6 were rejected.

For peptide and protein identification spectra files (RAW-files) were searched using MaxQuant (version 1.6.14.0) with default parameters for non-

carbamidomethylated peptides[71]. Protein sequences for database search were obtained from Uniprot (Proteome ID UP000001425, *Synechocystis* sp. PCC 6803, downloaded February 1, 2021). Peptides and proteins were filtered to satisfy a false discovery rate of 1%.

**EPR sample preparation**. EPR samples were prepared in an MBraun UniLab-plus glove box. 100 μL ~10 μM *Synechocystis* PS-CI in a 4.0 mm OD quartz EPR tube (Wilmad), and 10 μL ~20 μM *T. elongatus* PS-CI in a 1.6 mm OD Suprasil quartz EPR tube (Goss Scientific), were reduced in 20 mM sodium dithionite (Sigma, in Tris pH 9.5) for fully reduced spectra. The fully reduced samples were used for further pulsed EPR investigations (Fig. 2).

Potentiometric titrations (Fig. 3) on ~10 μM protein were carried out as previously described[46]. PS-CI was reduced or oxidised with substoichiometric amounts of sodium dithionite or $K_3Fe(CN)_6$ (Sigma) under anaerobic conditions in an electrochemical glass cell equipped with a 4 °C water bath. Once equilibrated under nitrogen while stirring, 30 μM of the redox mediators methylene blue, indigotrisulfonate, indigodisulfonate, anthraquinone-2-sulfonate, benzyl viologen and methyl viologen (Sigma Aldrich) were added. The potential was measured using an Ag/AgCl mini-reference electrode (DRI-REF-2, World Precision Instruments) and a platinum working electrode (Scientific Glassblowing Service, University of Southampton; Pt from Goodfellow) and connected to an EmSTAT3 + potentiostat (PalmSens). Samples (~10 μL) were transferred to 1.6 mm OD Suprasil quartz EPR tubes (Goss Scientific) at the indicated potentials and flash-frozen in ethanol cooled from outside the glovebox by a dry ice acetone bath, before being transferred to liquid nitrogen. All reduction potentials are given relative to the potential of the SHE. The reference electrode potential was determined to be +201 mV vs. SHE using quinhydrone (Sigma Aldrich) as an external standard.

**CW EPR spectroscopy**. EPR measurements were performed using an X-band Bruker Elexsys E580 Spectrometer (Bruker BioSpinGmbH, Germany) equipped with a closed-cycle cryostat (Cryogenic Ltd, UK) using Xepr software. All *T. elongatus* and titration measurements were carried out in an X- band split-ring resonator (ER 4118X-MS2). The field was calibrated using a DPPH standard (Bruker). The *Synechocystis* fully reduced sample was measured using an ER 4118X-MD5 resonator. Baseline spectra from samples containing only buffer or oxidised PS-CI were used as background and subtracted from the CW spectra. Unless otherwise specified data were collected at 15 K, 2 mW microwave power, 100 kHz modulation frequency, 7 G modulation amplitude and 16 scans.

**Relaxation filtered pulsed EPR spectroscopy**. *T. elongatus* and *Synechocystis* samples were measured in an X-band split-ring (ER 4118X-MS2) and a dielectric ring (ER 4118X-MD5) resonators, respectively, both mounted on a modified standard EPR probe head containing a low-noise cryogenic preamplifier, which significantly enhances the EPR sensitivity[37]. Two-pulse echo-detected field sweeps (EDFS) were acquired with the pulse sequence $\pi/2-\tau-\pi-\tau-echo$. $T_2$ was determined by varying $\tau$; the signal intensity of the simulated clusters was fitted with a single exponential. The $T_1$-relaxation filtered EDFS were obtained using the sequence $\pi$ -$T_f-\pi/2-\tau-\pi-\tau-echo$, where $T_f$ denotes the filtering time. Unless otherwise stated, the fully reduced *T. elongatus* PS-CI sample was measured at 10 K with $\pi = 32$ ns, $\tau = 250$ ns, short repetition time (SRT) of 2.04 μs. *Synechocystis* PS-CI was measured at 10 K, $\pi = 32$ ns, $\tau = 250$ ns, SRT = 8.16 μs.

**DEER spectroscopy**. DEER measurements used a four pulse sequence, $\pi_A/2-\tau_1-\pi_A-(\tau_1 + t)-\pi_B-(\tau_2-t)-\pi_A-\tau_2-echo$[44], with detection pulses at frequency $\omega_A$, and a single pump pulse at frequency $\omega_B$, which moved within the refocused echo sequence by variation of time $t$, e.g. steps of 12 ns. Experimental parameters for each position recorded are detailed in Supplementary Table 5. Position frequencies within the EPR spectrum are annotated in Supplementary Fig. S3. An eight-step phase cycle was employed to remove unwanted echoes. Positions 1 to 5 were measured in ER 4118X-MS2, with $\tau_1 = 134$ ns and $\tau_2 = 1.28$ μs. Positions 6 to 9 were measured in ER 4118X-MS2 resonator equipped on a probe head containing a low-noise cryogenic preamplifier for increased sensitivity[37], with $\tau_1 = 400$ ns and $\tau_2 = 1$ μs. All experiments were conducted at 10 K. DEER spectra were normalised to the zero-time intensity.

**Simulation of EPR spectra and analysis**. CW and EDFS data were analysed and simulated with EasySpin esfit using Monte Carlo simulation in Matlab[72]. All spectra, which were consistent amongst different protein batches, were simulated using the parameters reported in Supplementary Table 3. 'Nernst plots' (Fig. 3) were generated based on the integrated area of the simulated signals (due to EPR signal overlap), plotted against the reduction potential of the samples, normalised to the maximum intensity signal resulting from the fully reduced sample. The one-electron Nernst equation was fitted to the experimental data points using the Matlab curve-fitting toolbox. The mid-point reduction potential error was calculated based on the 95% confidence intervals for the regression over the linear section of the Nernst curve. For details of DEER, trace simulations see Supplementary Note 2 and the associated supplementary figures and tables.

**Reporting summary**. Further information on research design is available in the Nature Research Reporting Summary linked to this article.

## Data availability
Source data are provided with this paper. The data generated in this study have been deposited in the Imperial College London Research Data Depository database under DOI: 10.14469/hpc/8656.

Accessions for proteins and protein subunits used in this work are as follows: *Thermosynechococcus elongatus* photosynthetic complex I, 6HUM [www.wwpdb.org/pdb?id = pdb_00006hum] and *NdhF1*, Q8DKX9 [www.uniprot.org/uniprot/Q8DKX9]; *Synechocystis* sp PCC 6803 Ndh-J, P19125 [www.uniprot.org/uniprot/P19125]; *Synechocystis* sp. PCC6803 proteome [www.uniprot.org/proteomes/UP000001425]. Source data are provided with this paper.

## Code availability
Matlab-based codes for the analysis of absorbance and first derivative EPR spectra are available (github/KHRichardson/Nat_comms_2021). The algorithm used for the orientation-selective DEER analysis is detailed in Lovett et al., PCCP, 2009, 11, 6840-6848. Matlab-based codes used for the analysis of the DEER data are available from the corresponding author (M.M.R.) upon request.

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

## Acknowledgements

K.H.R. thanks the London Interdisciplinary Doctoral Programme for a studentship. The EPSRC (EP/P510841/1 seed funding to M.M.R and G.T.H as part of global challenges award, EP/T031425/1 to the Centre for Pulse EPR at Imperial College, EP/L011972/1 to the Centre for Advanced ESR at the University of Oxford and EP/P510270/1 to M.Š.), BBRSC (BB/R004838/1 to G.T.H), John Fell Fund (0007019 to W.K.M.), DFG priority programme 2002 (NO 836/4-1 to M.M.N.) and Leverhulme Trust (RPG-2018-183 to M.M.R.) are gratefully acknowledged for funding.

## Author contributions

K.H.R. performed all data analysis and research except those specified below. G.T.H., M.M.R. and K.H.R. designed the research with assistance from J.J.W. and G.M. G.T.H. and M.M.R. directed the research. K.H.R. and J.T. purified PS-CI from *T. elongatus* cells grown by J.T. and M.M.N. A.M.E. cloned the NdhJ-His *Synechocystis* sp pcc 6803 strain. W.K.M. performed DEER experiments 1–5. M.H. performed the mass-spectrometry on PS-CI purified from *Synechocystis* sp pcc 6803. K.H.R. and M.S. measured DEER traces 6–9 and the relaxation filtered EPR using the HEMT probe designed by M.S. and J.J.L.M. K.H.R, M.M.R. and G.T.H. wrote the manuscript. All authors read and approved the final manuscript.

## Competing interests

The authors declare no competing interests.
