## [Peer Review File · Nature Communications]

Functional basis of electron transport within photosynthetic complex IREVIEWER COMMENTS

Reviewer #1 (Remarks to the Author):

PS-CI is structurally related to R-CI but accepts electrons from Fd rather than NADH. It has been unclear how the electron transport from Fd to PQ is coupled with proton translocation. In this study, the authors determined g values of three Fe-S centers (N0, N1 and N2) and their positions in the structures. They also determined reduction potentials of three clusters. N1 and N2 closely located the PQ reduction site have the similar relatively high reducing potentials, suggesting that they function as electron sinks to prevent the reverse reaction.

As emphasized in the text, it has been unclear how the electron transport in CI is coupled with proton translocation even in R-CI. But this manuscript is unlikely to clarify something on it, although the authors tried to discuss something. Introduction may be unsuccessful because it makes the impression of this manuscript slightly more negative. I understand that it is a basis for the further research to determine the reduction potentials but readers may expect some biological messages. I could not understand the technical background of the EPR analysis and its technical soundness. I also could not evaluate the impact of the technical breakthrough. I list some specific points.

- 1) Lines 221-222. It has been reported that binding of NdhV to the main complex may alter the mode of electron donor binding. Was the complex analyzed in this study fully equipped with this subunit? Is the information in supplementary Tables 1 and 2?
- 2) It is interesting that N1 and N2 have the similar reduction potentials. It may be difficult to compare the situation with R-CI because the reduction potential of the center corresponding to N2 is unclear in R-CI. In general, however, the lack of the comparison with R-CI makes the manuscript less exciting.
- 3) Lines 48-49. The authors are probably discussing C4 photosynthesis. Specify it for the clarity. In C4 plants, PS-CI is more abundant in bundle sheath cells than mesophyll cells but its function is for the entire pathway of photosynthesis in a leaf rather than the cell-type specific function. All C4 plants are not crops. Many recent works have clarified that PS-CI is also important in C3 plants.
- 4) Lines 132-133. What is a semi-reduced cluster? Does it mean the pool is partially reduced?
- 5) Refs 1 and 31 are the same.

Reviewer #2 (Remarks to the Author):

In this manuscript the characterization of the iron-sulfur centers in photosynthetic complex I from *Thermosynechococcus elongatus* are characterized spectroscopically by EPR spectroscopy. Their redox potentials, their g-tensors and their relative positions have been determined by pulsed EPR spectroscopy. The results are compared with published data from respiratory complex I. This is very interesting and shows the potential of advanced EPR spectroscopy to unravel bioenergetics and structural aspects of these important electron transfer cofactors in such large protein complexes. Thus I believe that these findings are important enough to be published in Nature Communications.

Having said that I think that the manuscript needs (especially for the supporting information) a major revision. Many details of the experimental results and of the parameters entering the simulations shown in Figure 2 of the manuscript are missing, which makes it impossible to check the conclusions drawn. The authors stress that relaxation filtering was crucial to disentangle the spectra of the three iron-sulfur clusters, but do not give any quantitative numbers for these relaxation times. Do they find orientation independent relaxation times for each of the three clusters? How far are they separated? These numbers, which are important to perform the fitting as for example shown in Figure S1 should be added to the table where the g-values are given. Secondly, the authors state that the centers are only partially reduced under some experimental conditions. Here it should be stated in what ratios the intensities of the three centers were taken into account; for example for the DEER trace simulations. The same is true for the simulation of the orientation selective DEER traces. Only the distances between the iron-sulfur centers are reported, but the angle between the two connecting vectors as well as the orientation of the g-tensor axis of the individual centers play an important role. Without these missing information the conclusions can neither be checked nor verified. In Figure S4 the DEER data for the different pump/probe positions are reported as 'experimental traces' but it should be stated if intermolecular background decay was already divided out. Also it should be stated if the simulations also fitted the experimental modulation depth or if these have been adjusted (by how much). Looking on the quality of the fits that would be very astonishing (but of course nice).

In the dipolar frequency formula in the SI there is a typo: the B there should be a lower case index and not a large B and the angle θ is not defined.

Only if this additional information is given in the SI it will be possible for experts in the field of EPR spectroscopy to follow the important conclusions presented in the main manuscript.

Reviewer #3 (Remarks to the Author):

Review: Functional basis of electron transport within photosynthetic complex I

This is an interesting study in which the authors disentangle the electron pathway of Photosynthetic Complex I (PS-CI) by measurement of the iron-sulphur clusters cofactors reduction potentials with assignment based on continuous-wave and pulsed EPR experiments. A well-documented SI section is also provided.

The work is well-performed and makes a useful contribution to the field. The authors of Nature Communications should be very interested in this study and I recommend for publication.

I offer several points for the authors to address:

- 1) Page 6, line 92: The authors write that information on the FeS reduction potentials is already known but they do not specifically state what information??
- 2) Page 7, Figure 2. No axis label is given for the DEER traces. A label should be provided. Display in this way gives no indication of DEER modulation depth. A comment on the magnitude of the modulation depth should be given, either in the figure legends or SI. This is important as it gives an indication of the amount of spin pairs present in the sample as well as excitation bandwidth.
- 3) page 8, line 164: The authors give the impression that ref 45 is the origin of DEER. The authors should cite the original PELDOR papers or note the four-pulse version.
- 4) page 8, line 174: The authors fail to justify the claim of two models. Why such certainty when there are seven clusters and only three appear EPR active in their experiments????? What is the error in DEER distance possibly helping govern the choice of two models?
- 5) page 9: Figure 3B. What is the original of the unsimulated signal in the inset? Is this the quinone? This should be mentioned in the figure legend.
- 6) page 19. It seems relaxation filtering was not used for the DEER experiments. Why not? Does this mean the DEER traces contain information from 3 spin centres?

SI:

- 7) Page 18. Figure S4 reveals some cherry-picking of the data shown in the main text. Why are simulation traces 2 and 6 not matching the oscillation? Is the discrepancy caused by error in the interspin vector angle between the clusters? More discussion of sources of error would help the justification.

Reviewer #4 (Remarks to the Author):

To elucidate the electron pathway between the FeS clusters in the PS-CI is interesting. The authors make an attempt to do that to study the distance and redox potentials of these FeS clusters. These results is worth publishing in Nature Communication after the revision, since the obtained distance of 26 Å might have other different function during the electron transfer. The following questions should be stressed in the revision.

1) In R-CI, the final acceptor UQ is ~30 Å far from N2 cluster. What the possible distance between the N2 and PQ is in PS-CI ? This is not mentioned in the present manuscript.

2) The potential of the reduced Fd is -400 mV. The found redox potentials (-230 mV) of N2 and N1 are similar to each other, meaning that electron transfer from the final donor Fd to N2 is as favorable as to N1. Does this means that N1 or N2 can be reduced by N0 or Fd independently ? If this hold true, N0 is a electron shuttle between Fd and N1/N2, regulating by the conformational change as the iron-sulfur protein in Cyt bc1 (Complex III). Therefore, the assignment of 26 Å from DEER to the distance between N1 and N2 is more favorable, which blocks the electron transfer between them(the optimal distance for the electron tunnel is less than ~14 Å). The electron transfer to PQ is either via N1 or N2 (likewise two pathways in Photosystem I), and this ensure the efficiency of cyclic electron flow. As a result, N0, N1 and N2 are not linear as proposed herein. The flexibility of N0-subunit also can deteriorate its EPR and DEER signals.

3) In Fig 1a, it is puzzled where the intermembrane space is.

RESPONSE TO REFEREES

We are delighted that overall our work was well-received, and we would like to thank all reviewers for their comments, which enabled us to improve the manuscript and the supplementary information. Changes to the manuscript and the supplementary information are highlighted in red.

Reviewer #1:

PS-CI is structurally related to R-CI but accepts electrons from Fd rather than NADH. It has been unclear how the electron transport from Fd to PQ is coupled with proton translocation. In this study, the authors determined g values of three Fe-S centers (N0, N1 and N2) and their positions in the structures. They also determined reduction potentials of three clusters. N1 and N2 closely located the PQ reduction site have the similar relatively high reducing potentials, suggesting that they function as electron sinks to prevent the reverse reaction.

As emphasized in the text, it has been unclear how the electron transport in CI is coupled with proton translocation even in R-CI. But this manuscript is unlikely to clarify something on it, although the authors tried to discuss something. Introduction may be unsuccessful because it makes the impression of this manuscript slightly more negative. I understand that it is a basis for the further research to determine the reduction potentials but readers may expect some biological messages. I could not understand the technical background of the EPR analysis and its technical soundness. I also could not evaluate the impact of the technical breakthrough. I list some specific points.

We agree that our findings do not allow direct insights into the coupling mechanism, although they reveal necessary underpinning information (namely the reduction potentials of the clusters), without which it is difficult to understand the function of the complex. To reflect this, we now emphasise in the introduction the importance of understanding electron transfer in the first instance, which our work makes possible.

1) Lines 221-222. It has been reported that binding of NdhV to the main complex may alter the mode of electron donor binding. Was the complex analyzed in this study fully equipped with this subunit? Is the information in supplementary Tables 1 and 2?

We do detect NdhV by mass spectrometry in the *T. elongatus* PS-CI, but not in *Synechocystis* (Supplementary Tables 1 and 2). However, the mass spectrometry data do not provide a quantitative analysis. Our complex was prepared by the same method used previously to solve the structure by Cryo-electron microscopy (Schuller et al., Science, 2019, 260, 257-260), in which NdhV was not observed. NdhV is also only loosely associated with PS-CI and we therefore believe that only a sub-stoichiometric proportion of the PS-CI analysed here will have NdhV associated.

2) It is interesting that N1 and N2 have the similar reduction potentials. It may be difficult to compare the situation with R-CI because the reduction potential of the center corresponding to N2 is unclear in R-CI. In general, however, the lack of the comparison with R-CI makes the manuscript less exciting.

With the exception of R-CI from *Thermo thermophilus* where the N2 cluster has not been observed by EPR, the reduction potential of N2 in R-CI has been determined in many different species (as summarised in Table 1 in Hirst & Roessler, BBA, 2016, 1857, 7, 872-883). N2 acts as an electron sink in R-CI in almost all species, except for *E. coli*, where the relatively positive reduction potential may be the result of the enzyme adapting to using menaquinone under anaerobic conditions. We note that the absolute values for the redox potentials vary between species (and depend on pH), making direct comparison with PS-CI difficult. However, it is possible to compare the relative difference in potentials between clusters in PS-CI and R-CI.

The potential of the cluster homologous to N1 is unknown in R-CI, however it has been shown to remain oxidised by Moessbauer spectroscopy (Bridges et al., *Biochemistry*, 2012, 51, 1, 149-158) suggesting its reduction potential is below the potential that can be reached with NADH or even Europium (see Reda et al., *Biochemistry* 2008, 47, 34, 8885–8893).

Comparisons with respiratory complex I have now been included in the text (p. 8, line 182-187).

3) Lines 48-49. The authors are probably discussing C4 photosynthesis. Specify it for the clarity. In C4 plants, PS-CI is more abundant in bundle sheath cells than mesophyll cells but its function is for the entire pathway of photosynthesis in a leaf rather than the cell-type specific function. All C4 plants are not crops. Many recent works have clarified that PS-CI is also important in C3 plants.

We thank the reviewer for pointing out that this could be more clearly expressed. We now specifically describe the increased abundance of PS-CI in C4 photosynthesis and its importance in specific conditions for C3 plants (p. 2-3, line 46-49).

4) Lines 132-133. What is a semi-reduced cluster? Does it mean the pool is partially reduced?

We have now changed 'semi-reduced' to 'partially reduced' and have clarified that this means that approximately half of the clusters in the enzyme that give rise to this third set of EPR signals are reduced (p. 5-6 lines 123-124).

5) Refs 1 and 31 are the same.

We thank the reviewer for noticing this; the duplicate reference has been removed.

Reviewer #2:

In this manuscript the characterization of the iron-sulfur centers in photosynthetic complex I from *Thermosynechococcus elongatus* are characterized spectroscopically by EPR spectroscopy. Their redox potentials, their g-tensors and their relative positions have been determined by pulsed EPR spectroscopy. The results are compared with published data from respiratory complex I. This is very interesting and shows the potential of advanced EPR spectroscopy to unravel bioenergetics and structural aspects of these important electron transfer cofactors in such large protein complexes. Thus I believe that these findings are important enough to be published in Nature Communications.

We thank the reviewer for these positive comments.

Having said that I think that the manuscript needs (especially for the supporting information) a major revision. Many details of the experimental results and of the parameters entering the simulations shown in Figure 2 of the manuscript are missing, which makes it impossible to check the conclusions drawn.

We apologise that we did not originally succeed in presenting all the information in the best possible way and we hope that our revised version addresses this. We do think that it was possible to check our conclusions as the simulation model for the orientation selective DEER traces is described in detail in Lovett et al., PCCP, 2009, 11, 6840-6848 as well as in Roessler et al., PNAS, 2010, 107, 1930-1935 (as stated in Supplementary Note 2). However, we agree that it is helpful for the readers of our paper to have more detail in our supplementary information, and these have now been included. We have also edited the legends of many of the supplementary tables and figures to improve clarity, included additional cross-references, and added two supplementary figures to better explain the simulation model (details below).

Figure 2 contains a large amount of spectroscopic data. Whilst we have now added more detail on the CW EPR measurement parameters, adding all of the parameters for pulsed experiments in addition to the simulation parameters would make the legend incomprehensibly long. We now refer to supplementary figures and notes in which these parameters are detailed.

The authors stress that relaxation filtering was crucial to disentangle the spectra of the three iron-sulfur clusters, but do not give any quantitative numbers for these relaxation times. Do they find orientation independent relaxation times for each of the three clusters? How far are they separated? These numbers, which are important to perform the fitting as for example shown in Figure S1 should be added to the table where the g -values are given.

As all three clusters are [4Fe-4S] clusters, their relaxation properties are unlikely to differ very significantly and indeed under no experimental conditions could we truly separate a single species. Hence inversion recovery of the spin echo still results in a mixture of contributions from the three clusters. However, using a model of the g values and strain we can comment on observed relaxation times. Quantitative numbers of the simulated species from inversion recovery for T_1 and two pulse EDFS for T_2 fit with a single exponential have been added to the manuscript (Supplementary Page 3 and Table S3). Inversion recovery does indicate some orientation dependent relaxation, however as we are unable to quantify that for individual clusters, the g strain was allowed to vary when fitting the individual EDFS to compensate for this. This is now discussed in Supplementary Note 1: the g strains used for the final fits presented in the manuscript (Table S3) were optimal for the CW spectra (Fig. 2a) and EDFS without T_1 filtering (Fig. 2b and Fig. S3).

Secondly, the authors state that the centers are only partially reduced under some experimental conditions. Here it should be stated in what ratios the intensities of the three centers were taken into account; for example for the DEER trace simulations. The same is true for the simulation of the orientation selective DEER traces.

The relative weightings of the clusters for the DEER trace simulation have now been added to Supplementary Table 3. In Figure 2 and Supplementary Figures 1, 2, 3b and 7 the ratios were already stated.

Only the distances between the iron-sulfur centers are reported, but the angle between the two connecting vectors as well as the orientation of the g-tensor axis of the individual centers play an important role. Without these missing information the conclusions can neither be checked nor verified.

We apologise for not originally including this key information in the text. The angles and g-tensor axis for the best-fit shown in Supplementary Figure 4 were checked but not included in the supplementary; this has been rectified by the addition of Supplementary Figure 6.

To further ascertain our conclusions, we have also re-calculated all the DEER traces for another set of spin projection factors suitable for ferredoxin-type FeS clusters (as found in PS-CI), $k_{1,2}=+1.85$ $k_{3,4}=-1.35$. These were found to fit the data slightly less well compared to the $k_{1,2}=+1.17$ $k_{3,4}=-0.67$ set and thus the corresponding best fits for the latter used in Figure 2 and Figure S4. We also now include a set of best-fit simulations (Fig S5) where SPFs and g tensors are a free variable between all the traces. Although this is physically not meaningful, it does corroborate our conclusion, as even in this case model A does not fit.

In Figure S4 the DEER data for the different pump/probe positions are reported as ‘experimental traces’ but it should be stated if intermolecular background decay was already divided out.

Intermolecular decay was removed by subtracting a linear baseline, this has been clarified in the Figure S4 legend.

Also it should be stated if the simulations also fitted the experimental modulation depth or if these have been adjusted (by how much). Looking on the quality of the fits that would be very astonishing (but of course nice).

The total simulations are weighed to the estimated number of spins at the detection position for each cluster (see Table S3). This has been further clarified in Supplementary Note 2 with the addition of Equations 7 & 8. Modulation depths scale bars have now been added to Figure 2c,d and Figure S4. Besides this weighting, the modulation depths are those calculated by the OriDEERSim programme, i.e. the fraction of B-spin signal at a given orientation that is flipped by the pump π pulse (see Lovett et al., PCCP, 2009, 11, 6840-6848 for details).

In the dipolar frequency formula in the SI there is a typo: the B there should be a lower case index and not a large B and the angle Ψ is not defined.

We thank the reviewer for noticing. This has been rectified and Supplementary Figure 6 has been added to aid visualisation.

Only if this additional information is given in the SI it will be possible for experts in the field of EPR spectroscopy to follow the important conclusions presented in the main manuscript.

We would like to thank the reviewer for the careful reading of our manuscript. We hope that our substantially revised supporting information, as well as some changes in the main manuscript, now address all the points raised and fully justify our conclusions.

Reviewer #3:

Review: Functional basis of electron transport within photosynthetic complex I

This is an interesting study in which the authors disentangle the electron pathway of Photosynthetic Complex I (PS-CI) by measurement of the iron-sulphur clusters cofactors reduction potentials with assignment based on continuous-wave and pulsed EPR experiments. A well-documented SI section is also provided.

The work is well-performed and makes a useful contribution to the field. The authors of Nature Communications should be very interested in this study and I recommend for publication.

We thank the reviewer for these positive comments on our work.

I offer several points for the authors to address:

1) Page 6, line 92: The authors write that information on the FeS reduction potentials is already known but they do not specifically state what information??

A species with the same g values as N2 was observed in Nostoc thylakoid membranes, with a redox potential of $-270 \text{ mV} \pm 25 \text{ mV}$ (as stated on page 5). At the time the authors of this study were not working with the isolated enzyme and hence could not determine the origin of this signal.

2) Page 7, Figure 2. No axis label is given for the DEER traces. A label should be provided. Display in this way gives no indication of DEER modulation depth. A comment on the magnitude of the modulation depth should be given, either in the figure legends or SI. This is important as it gives an indication of the amount of spin pairs present in the sample as well as excitation bandwidth.

Modulation depths have been added to Figure 2 and Supplementary Figures 4 and 5. We agree with the reviewer that the modulation depth is an important indicator and have added a comment and further clarification to Supplementary Note 2.

3) page 8, line 164: The authors give the impression that ref 45 is the origin of DEER. The authors should cite the original PELDOR papers or note the four-pulse version.

A citation for the original PELDOR paper has been added.

4) page 8, line 174: The authors fail to justify the claim of two models. Why such certainty when there are seven clusters and only three appear EPR active in their experiments????? What is the error in DEER distance possibly helping govern the choice of two models?

There are only three clusters present in PS-CI. Perhaps there was a confusion because R-CI contains 8 (or even 9 in some species) Fe-S clusters. However, we only investigate PS-CI here. Because N2 is structurally and spectroscopically almost identical to R-CI, there are only two possibilities for the location of N1 and N0 and hence only two models.

5) page 9: Figure 3B. What is the original of the unsimulated signal in the inset? Is this the quinone? This should be mentioned in the figure legend.

The $g \sim 2$ signal likely arises from the redox mediators added to the titration. We cannot however definitively prove its origin. A note has been added to the figure 3 legend.

6) page 19. It seems relaxation filtering was not used for the DEER experiments. Why not? Does this mean the DEER traces contain information from 3 spin centres?

Adding a relaxation filter would have extended the time of the experiments and further decreased the signal to noise ratio (which is already a challenge, hence the use of the cryogenic preamplifier at the extreme field positions where few spins are excited).

The traces do therefore contain information from 3 spin centres as stated on p. 26, 578-579. The simulations for the DEER traces shown indeed only take into account two spin centres. We did also conduct simulations with all three spin centres; however no significant difference in the fits was observed when all three potential interactions were taken into account in the simulations. This is because the middle cluster is $<15 \text{ \AA}$ from the other two clusters, so that the modulation frequency is too high to be observed with the DEER pulse sequence employed (see also Supplementary Note 2, which now contains further details), and mostly lost in the deadtime for the pulse lengths used.

SI:

7) Page 18. Figure S4 reveals some cherry-picking of the data shown in the main text. Why are simulation traces 2 and 6 not matching the oscillation? Is the discrepancy caused by error in the interspin vector angle between the clusters? More discussion of sources of error would help the justification.

The best-fitting calculated DEER traces were selected based on least squares fitting with the experimental data. The model for the best-fit g -tensor and spin projection factor assignment for all simulations in Supplementary Figure 4 and Figure 2 in the main paper has now been added as Supplementary Figure 6.

The local spin model used in the simulations is an approximation with associated errors. Further discussion of errors has been added to Supplementary Note 2 (page 5). The assignment of the spin projection factors, and their exact values, will affect the DEER frequency. Moreover, in the model used, the co-ordinates for the Fe ions were taken from a 3.3 \AA resolution cryo-EM structure of PS-CI (PDB 6HUM, Schuller et al., Science, 2019, 260, 257-260). Uncertainty in the coordinates associated with the resolution of the structure are thus an additional source of error in the simulated DEER traces.

Reviewer #4:

To elucidate the electron pathway between the FeS clusters in the PS-CI is interesting. The authors make an attempt to do that to study the distance and redox potentials of these FeS clusters. These results is worth publishing in Nature Communication after the revision, since the obtained distance of 26 Å might have other different function during the electron transfer. The following questions should be stressed in the revision.

1) In R-CI, the final acceptor UQ is ~30 Å far from N2 cluster. What the possible distance between the N2 and PQ is in PS-CI ? This is not mentioned in the present manuscript.

Based on density observed in a crystal structure of R-CI and the structure of R-CI with a quinone binding site inhibitor bound, UQ is approximately ~12Å from N2 (Baradaran et al, Nature, 2013, 494, 7438, 443-448, Bridges et al, Nat. Commun., 2020, 11, 5261). Given the structural similarity of PS-CI and R-CI in this region we assume the distance would be similar in PS-CI, and efficient for electron transfer. The estimated distance and analogy with R-CI has been added to the discussion (p. 9 line 215-6).

2) The potential of the reduced Fd is -400 mV. The found redox potentials (-230 mV) of N2 and N1 are similar to each other, meaning that electron transfer from the final donor Fd to N2 is as favorable as to N1. Does this means that N1 or N2 can be reduced by N0 or Fd independently ? If this hold true, N0 is a electron shuttle between Fd and N1/N2, regulating by the conformational change as the iron-sulfur protein in Cyt bc1 (Complex III). Therefore, the assignment of 26 Å from DEER to the distance between N1 and N2 is more favorable, which blocks the electron transfer between them(the optimal distance for the electron tunnel is less than ~14 Å). The electron transfer to PQ is either via N1 or N2 (likewise two pathways in Photosystem I), and this ensure the efficiency of cyclic electron flow. As a result, N0, N1 and N2 are not linear as proposed herein. The flexibility of N0-subunit also can deteriorate its EPR and DEER signals.

There are two structures published of PS-CI with Fd bound, showing it binds to the top of the complex ~10Å next to N0 (Zhang et al., Nat. Commun., 2020, 11,888; Pan et al., Nat. Commun., 2020, 11, 610). The only pathway between Fd to PQ with distances efficient for transfer, below 14Å, would be the linear pathway revealed by the structures. In R-CI this has been modelled by DFT indicating efficient electron tunnelling through the linear pathway, with electrons entering the chain at the top and stabilised preferentially on N2 (Hayashi & Stuchebrukhov, PNAS, 2010, 107,45,19157-19162). All electrons from Fd are therefore almost certain to pass through the N0 cluster. Although it is probable that there are conformational changes on Fd binding, there is currently no evidence to support the large structural rearrangement that would be necessary to bring Fd within 14Å of N1 or N2, or bring N1 to the quinone binding site. The rearrangement in Cyt bc1 mentioned by the reviewer is facilitated by the flexible subunit in which the 2Fe-2S cluster is bound. N2 is coordinated by NdhK and sits buried at the interface of three subunits, N1 and N0 are both coordinated by NdhI, resulting in a relatively rigid structure. In summary, it is difficult to envisage how large conformational changes, and hence non-linear electron transfer, could take place while maintaining the three 4Fe-4S clusters intact in the complex.

We agree that the EPR signal of N0 may be affected by Fd binding, and this is indeed mentioned in the manuscript (p. 8, line 197-198) in the context of the low reduction potential we deduce for this cluster. The broad lines could be due to spin diffusion, as corroborated by the short T_2 observed for N0, as well as conformational inhomogeneity due to flexibility.

3) In Fig 1a, it is puzzled where the intermembrane space is.

Both mitochondria and thylakoids have bilayers with the intermembrane space between them. For clarity the labels in Figure 1 have been expanded.

REVIEWERS' COMMENTS

Reviewer #1 (Remarks to the Author):

The authors responded to my concerns appropriately. If other reviewers are satisfied by their comments on the technical points, I would also recommend the publication.

Reviewer #2 (Remarks to the Author):

The authors revised their manuscript taking all the comments and suggestions of all reviewers into account. I am perfectly happy with this revised manuscript and recommend publication in Nature Communication now.

Reviewer #3 (Remarks to the Author):

The authors have successfully addressed the feedback.

Reviewer #4 (Remarks to the Author):

Since the author have improved the manuscript, I have no more comments. The revision is suitable for publication.